# PMEPA1/TMEPAI Is a Unique Tumorigenic Activator of AKT Promoting Proteasomal Degradation of PHLPP1 in Triple-Negative Breast Cancer Cells

**DOI:** 10.3390/cancers13194934

**Published:** 2021-09-30

**Authors:** Md. Anwarul Haque, Mohammed Abdelaziz, Meidi Utami Puteri, Thanh Thao Vo Nguyen, Kosei Kudo, Yukihide Watanabe, Mitsuyasu Kato

**Affiliations:** 1Department of Experimental Pathology, Graduate School of Comprehensive Human Sciences, Faculty of Medicine, University of Tsukuba, Tsukuba 305-8575, Japan; anwarulhaque@ru.ac.bd (M.A.H.); mohamed_mohamed1@med.sohag.edu.eg (M.A.); meidi.utami@md.tsukuba.ac.jp (M.U.P.); vntthao.snn@tphcm.gov.vn (T.T.V.N.); kosei-kudo.fh@cmic.co.jp (K.K.); mit-kato@md.tsukuba.ac.jp (M.K.); 2Department of Pharmacy, Faculty of Science, University of Rajshahi, Rajshahi 6205, Bangladesh; 3Department of Pathology, Faculty of Medicine, Sohag University, Sohag 82524, Egypt; 4Department of Medical Biotechnology, Biotechnology Center of Ho Chi Minh City, Ho Chi Minh City 700000, Vietnam; 5Transborder Medical Research Center, Faculty of Medicine, University of Tsukuba, Tsukuba 305-8575, Japan

**Keywords:** TMEPAI, TNBC, AKT signaling, PHLPP1, ubiquitination

## Abstract

**Simple Summary:**

TMEPAI/PMEPA1 is known to be highly expressed in various types of cancer, including triple-negative breast cancer (TNBC). Here, we found that TMEPAI-knockout (KO) in TNBC cells showed less tumorigenic abilities and upregulated PHLPP1, which dephosphorylates AKT at Ser473. Additionally, PHLPP1 depletion in TMEPAI-KO cells partially but significantly recovered AKT Ser473 phosphorylation, as well as in vitro and in vivo tumorigenic activities. Moreover, we demonstrated that TMEPAI PPxY (PY) motifs are essential for binding to NEDD4-2, an E3 ubiquitin ligase, and the complex formation of PHLPP1 with NEDD4-2, which leads to the polyubiquitination and proteasomal degradation of PHLPP1. These findings indicate that the PY motifs of TMEPAI induce the degradation of PHLPP1 and maintain AKT Ser473 phosphorylation at high levels to enhance the tumorigenic potentiality of TNBC. Therefore, the putative oncogenic role of TMEPAI may lead to developing TMEPAI based targeted therapy and/or diagnosis of TNBC in future.

**Abstract:**

Transmembrane prostate androgen-induced protein (TMEPAI), also known as PMEPA1, is highly expressed in many types of cancer and promotes oncogenic abilities. However, the mechanisms whereby TMEPAI facilitates tumorigenesis are not fully understood. We previously established TMEPAI-knockout (KO) cells from human triple-negative breast cancer (TNBC) cell lines and found that TMEPAI-KO cells showed reduced tumorigenic abilities. Here, we report that TMEPAI-KO cells upregulated the expression of pleckstrin homology (PH) domain and leucine-rich repeat protein phosphatase 1 (PHLPP1) and suppressed AKT Ser473 phosphorylation, which was consistent with TCGA dataset analysis. Additionally, the knockdown (KD) of PHLPP1 in TMEPAI-KO cells partially but significantly rescued AKT Ser473 phosphorylation, as well as in vitro and in vivo tumorigenic activities, thus showing that TMEPAI functions as an oncogenic protein through the regulation of PHLPP1 subsequent to AKT activation. Furthermore, we demonstrated that TMEPAI PPxY (PY) motifs are essential for binding to NEDD4-2, an E3 ubiquitin ligase, and PHLPP1-downregulatory ability. Moreover, TMEPAI enhanced the complex formation of PHLPP1 with NEDD4-2 and PHLPP1 polyubiquitination, which leads to its proteasomal degradation. These findings indicate that the PY motifs of TMEPAI suppress the amount of PHLPP1 and maintain AKT Ser473 phosphorylation at high levels to enhance the tumorigenic potentiality of TNBC.

## 1. Introduction

Breast cancer (BC) is a major cause of mortality among cancer-associated deaths in women all over the world. It is estimated that BC is diagnosed in about 2.1 million women each year [1]. Although significant improvements have been achieved in multidisciplinary BC treatment, including surgery, radiotherapy, chemotherapy, and immunotherapy, triple-negative breast cancer (TNBC), an aggressive subgroup of BC, remains hindered by many limitations, such as resistance to conventional chemotherapies and a lack of reliable biomarkers [2]. Therefore, identification of novel TNBC therapeutic targets and the molecular mechanisms underlying the therapeutic targets is urgently needed to develop predictive biomarkers for targeted therapies.

Signaling pathways activated by growth factors (GF), TGF-β, Notch, Hippo, and Wnt are involved in BC progression as they subsequently impact cell proliferation, survival, migration, differentiation, and apoptosis [3,4]. Among them, the phosphatidylinositol 3-kinase (PI3K)/AKT pathway plays a crucial role in the regulation of the cell proliferation, survival, and biological characteristics of both normal and malignant cells [5]. To activate PI3K signaling, receptor tyrosine kinases (RTKs) and other cell surface receptors engage with extracellular ligands and recruit PI3K to plasma membrane-anchored receptors for the production of phosphatidylinositol (3,4,5) trisphosphate [PI(3,4,5)P3], which further binds with multiple downstream effectors including phosphoinositide-dependent kinase (PDK) and AKT to recruit them at the membrane [6,7]. Then, PDK phosphorylates AKT at the activation loop (Thr308) [6], whereas the mammalian target of rapamycin (mTOR) complex 2 (mTORC2) actively localizes at the plasma membrane and endosomal vesicles, further phosphorylating AKT at the hydrophobic motif (Ser473) [8,9]. Phosphorylation of AKT at Thr308 contributes partially to the activation (~10%); its full activation requires phosphorylation at Ser473 [10,11]. Since the PI3K/AKT signal plays a pivotal role in many physiological events, the signal is also strictly regulated by negative regulatory molecules. The phosphatase and tensin homologue (PTEN), a well-known tumor suppressor, antagonizes the function of PI3K and inhibits the activation of AKT, particularly on Thr308. Moreover, PHLPP1 specifically dephosphorylates AKT at Ser473 [12,13,14,15,16]. In malignant cells, PI3K/AKT signaling is frequently activated owing to hyperactivation of RTKs, activating mutations of PI3K, or loss/inactivation of PTEN [12]. However, little is known about the dysregulation of the AKT Ser473 phosphorylation status in cancer cells. Several reports have indicated that PHLPP1 is downregulated in metastatic and aggressive breast cancer cells when compared with the levels in normal cells [17,18], but the mechanism of this downregulation of PHLPP1 in these cancer cells is not well studied.

TMEPAI is a type-Ib transmembrane protein, originally identified as an androgen-induced protein [19]. It is also known as prostate transmembrane protein, androgen-induced 1 (PMEPA1) or solid tumor-associated gene 1 (STAG1). TMEPAI protein has a short extracellular domain, followed by a transmembrane domain and a long intracellular domain that contains a Smad interaction motif (SIM) and two PY motifs [20]. In cancer cells, many signaling pathways are aberrated, which leads to uncontrolled cell proliferation, survival, motility, and metabolism. This dysregulation of the signaling pathways is due to the overexpression or constitutive activation of many oncogenic proteins, of which TMEPAI is one. TMEPAI is highly and constitutively expressed in many cancers including breast, lung, colorectal, ovary, and renal cancers [19,21,22,23], and the transcriptional induction is regulated by transforming growth factor-β (TGF-β), cooperatively with other oncogenic signaling [24,25,26]. Previous studies have shown that depletion of TMEPAI reduces tumorigenic activities in many types of cancer cells through the regulation of intracellular signaling pathways [27,28,29]. Singha et al. first demonstrated that TMEPAI downregulates PTEN to promote PI3K/AKT signaling, but the activation mechanism of AKT S473 phosphorylation remains unclear [12].

In this study, we investigate the roles of TMEPAI in TNBC and discover that the PY motifs of TMEPAI promote PI3K/AKT signaling by suppressing PHLPP1. Knockdown (KD) of PHLPP1 from TMEPAI-KO cells can restore AKT phosphorylation and tumorigenic ability. We also demonstrate here that TMEPAI, through its PY motifs, interacts with NEDD4-2 and facilitates PHLPP1 degradation. Therefore, we believe our study has identified a novel molecular function of TMEPAI, which is essential for full activation of AKT in TNBC cells together with genomic or epigenetic alterations in growth factor receptor, *PI3K* and *PTEN* genes and will encourage the design of TMEPAI-based targeted therapy for the treatment of TNBC in the future.

## 2. Materials and Methods

### 2.1. Analysis of RNA Sequencing Data in TCGA-Invasive Breast Cancer (BRCA) Cases

The RNAseq data of TCGA invasive breast cancer cases (BRCA) were downloaded from FireBrowse. Available online: http://firebrowse.org/ (accessed on 22 march 2020). Based on RESM-normalized RNAseq values of TMEPAI, a total of 1212 RNAseq samples (from 1093 BRCA cases) were arranged in descending order where samples in the top quartile were labeled TMEPAI.High while samples in the bottom quartile were labeled TMEPAI.Low (303 samples each).

Gene set enrichment analysis (GSEA) was performed by GSEA 4.1.0 software [30] to identify the enriched biological pathways in TMEPAI.High vs. TMEPAI.Low samples, using 587 gene sets derived from the WikiPathways subset of canonical pathways (CP) in the curated gene sets (C2) of MSigDB 7.2 [31]. In the gene set enrichment analysis (GSEA), the enrichment statistic and metric for ranking genes were set to Weighted and Signal2Noise, respectively. The statistical significance of the enrichment scores was assessed against 1000 gene set permutations using the analysis phenotype of TMEPAI.High vs. TMEPAI.Low. Gene sets with FDR *q*-value < 0.05 were considered significant.

For the association analysis of TMEPAI expression levels and phosphorylation of AKT, we used RNAseq and RPPA (Reverse Phase Protein Array) data files from FireBrowse to filter 771 cases in BRCA that had matched single RNAseq and single RPPA samples based on the corresponding TCGA barcodes. We arranged the samples in the filtered list in descending order according to RSEM-normalized RNAseq values of TMEPAI, where samples in the top quartile were labeled TMEPAI.High and samples in the bottom quartile were labeled TMEPAI.Low (193 samples each). The difference between TMEPAI.High and TMEPAI.Low groups regarding the level of AKT phosphorylation on Thr308 or on Ser473 was evaluated by Mann–Whitney test in GraphPad Prism 7 (GraphPad Softwere, San Diego, CA, USA) using ranks of RPPA values of AKT phosphorylation on Thr308 or on Ser473, respectively, where *p*-value < 0.05 was considered statistically significant.

### 2.2. Cell Lines and Cell Culture

Triple-negative breast cancer cell lines (Hs578T, BT-549, and MDA-MB-231) and HEK-293T cells were obtained from the American Type Culture Collection (ATCC) and cultured in Dulbecco’s modified Eagle’s medium (DMEM, Wako, Osaka, Japan) supplemented with 10% fetal bovine serum, 100 U/mL penicillin G, 0.1 mg/mL streptomycin sulfate, and an additional supplement of 10 μg/mL insulin for Hs578T. The cells were incubated in a humidified atmosphere (5% CO_2_ at 37 °C). TMEPAI-KO cells from these cell lines were established previously using the CRISPR/Cas9 gene editing technique [28,29]. Two TMEPAI KO clones (using two independent gRNAs) from each of the cell lines were used in this study. Before beginning the experiments, we checked the cells for mycoplasma contamination.

### 2.3. Plasmid Construction

A double PY motif mutant (Y161A, Y232A) and a SIM mutant (P186A or 4A (PPNR^186-189^ to AAAA)) of TMEPAI isoform A were generated using a QuickChange site-directed mutagenesis kit (Stratagene, San Diego, CA, USA), and cloned into a pcDNA3.1/V5-HisA vector for transient expression and a CSII-CMV-MCS-IRES2-Bsd vector for lentiviral stable expression. A PHLPP1 cDNA clone (HGX006251) was purchased from the RIKEN BioResource Center, Tsukuba, Japan, and replaced into a pcDNA3 vector with/without a FLAG-tag. The NEDD4-2 expression plasmid was a kind gift from Dr T. Imamura (Department of molecular medicine for pathogenesis, Ehime University, Ehime, Japan) [32].

### 2.4. Knockdown of Target Genes

For transient silencing of PHLPP1, NEDD4-2, and SMURF1, small interference RNAs (Ambion validated siRNAs) were purchased from Thermo Fisher Scientific, Tokyo, Japan, and transfected into the cells by use of Lipofectamine 3000 (Promega, Madison, WI, USA) according to the manufacturer’s protocol and then incubated for 48 h. Stealth siRNA Negative Control Med GC was used as the control. After incubation, knockdown (KD) of the target proteins was confirmed by means of Western blot analysis. For generation of stable KD of PHLPP1, PHLPP1-targeting lentivirus pLKO.1-shRNA was constructed using the following oligonucleotide sequences: (shPHLPP1: Forward: 5′CCGGAAGATTGATCAGCCTTCTACACTCGAGTGTAGAAGGCTGATCAATCTTTTTTG3′, Reverse: 5′AATTCAAAAAAAGATTGATCAGCCTTCTACACTCGAGTGTAGAAGGCTGATCAATCTT3′). The original pLKO.1 vector was used for the control shRNA. The lentivirus solution was produced by co-transfecting a pLKO.1-shRNA-expressing vector, pPAX2, and pMD2.G, into HEK-293T cells, and the filtrated culture supernatant was then used for infection into MDA-MB-231 cells in combination with 8 µg/mL polybrene. Stable KD cell lines were selected by using 1 µg/mL puromycin. KD of PHLPP1 was confirmed by means of Western blot analysis.

### 2.5. Colony Formation Assay

Parental and TMEPAI-KO cells were seeded in 10-cm dishes (500 cells/dish) and incubated for 15 days. After incubation, the cells were washed twice in PBS, fixed in 4% paraformaldehyde, and stained with 0.05% crystal violet solution. Photographs of the stained colonies were taken, and the colonies were counted by use of ImageJ software version 1.8.0, National Institute of Health, Bethesda, MD, USA. The percentage of the colony formation rate was calculated using the following formula: percent of colony formation rate = (number of colonies after 15 days/number of initial seeding cells) × 100. The experiments were independently performed in triplicate.

### 2.6. Cell Proliferation Assay

Approximately 1 × 10^3^ cells from parental and TMEPAI-KO TNBC cells were seeded in each well of a 96-well plate and incubated for 0, 24, and 48 h. After indicated incubation, 20 μL/well of MTS reagent (Madison, WI, USA) was added and again incubated for 2 h. Following incubation, absorbance was measured at 490 nm.

### 2.7. Tumor Sphere Formation Assay

The trypsinized cells were washed twice with PBS and once with culture medium, and 1 × 10^3^ cells were resuspended in mammosphere-forming medium (DMEM/F12 supplemented with 20 ng/mL EGF, 20 ng/mL bFGF, 5 μg/mL insulin, 2% B-27 (Invitrogen, Waltham, MA, USA)), seeded into ultra-low attachment 6-well plates or poly-2-hydroxyethyl methacrylate-precoated dishes, and subsequently incubated for 10 days. If indicated, the number of tumor spheres was counted, and the diameter of the spheres was measured from triplicate wells.

### 2.8. Transwell Cell Migration Assay

The migratory ability of the parental and TMEPAI-KO cells was assayed in Transwell Chambers (Corning Inc., Corning, NY, USA) according to a published protocol [33], with some modifications. A cell suspension of approximately 3 × 10^4^ cells in 100 μL of the serum-free medium was seeded to the upper chamber. The lower chambers were filled with 500 μL of culture medium with 10% FBS and incubated for 12 h. After incubation, the cells that had traversed from the upper surface of the membrane of the chamber to the lower surface were washed twice with PBS, fixed with 4% paraformaldehyde, and stained with crystal violet solution. After two washes with Milli-Q, several microscopic fields were randomly selected, and the migrated cell numbers were counted using ImageJ software. The experiments were independently performed in triplicate.

### 2.9. Wound Healing Assay

The cells (approximately 5 × 10^5^) were seeded in each well of six-well plates and incubated for about 80% confluence. After that, a straight wound was made by scraping the cell monolayer using a pipet tip, and cell debris was removed by PBS washing. The photographs were taken initially at 0 h and after 24 h incubation under an Olympus microscope, and the wound closure area (A) was estimated through ImageJ software. The % of wound closure ability was calculated according to the following formula: Wound inhibition, A = [(A_0h_ − A_24h_)/A_0h_] × 100 (%)

### 2.10. TMEPAI Re-Expression

TMEPAI re-expression in the TMEPAI-KO BRCA cells was performed using a lentivirus expression system. Wild-type TMEPAI isoform A, a PY motif mutant, and a SIM mutant-expressing vector (CSII-CMV-MCS-IRES2-Bsd-TMEPAI) were transfected into HEK-293T cells along with pMDLg/pRRE, pRSV-Rev, and pMD2.G using FuGENE6 transfection reagent according to the manufacturer’s protocol. The lentiviral expression vectors produced in the culture supernatant were collected after 36, 48, 60, and 72 h of incubation. The filtered viral solution with 8 μg/mL polybrene was added to the culture medium of TMEPAI KO #5 (Hs578T) cells and infected for 24 h. Blasticidin S hydrochloride (8 μg/mL) was used to select the infected cells.

### 2.11. Western Blot Analysis

The cells (1 × 10^6^) were seeded in 10 cm dishes, incubated for 70–80% confluency, and then harvested. The collected cell pellets were lysed using TNE buffer (10 mM Tris pH 7.4, 150 mM NaCl, 1 mM EDTA, 1% NP-40) with a protease and phosphatase inhibitor cocktail. The lysates were cleared by removal of undissolved precipitates by centrifugation, and an equal amount of proteins from each sample was denatured and subjected to sodium dodecyl sulfate-polyacrylamide gel electrophoresis (SDS-PAGE) and electrotransferred onto a polyvinylidene difluoride (PVDF) membrane. The membrane was incubated for 40 min with 4% skimmed milk to block the nonspecific antibody from binding, and then incubated with respective primary antibodies overnight at 4 °C. The antibodies used in this study were anti-TMEPAI (homemade, [20], anti-PHLPP1 (Abcam, Cambridge, UK, ab71972), anti-AKT (CST, Dancers, MA, USA, 4056S), anti-AKT pSer473 (CST, 9271S), anti-NEDD4-2 (Abcam, ab46521), anti-SMURF1 (Abcam, 38866), anti-β-actin (Wako, 010-27841), anti-FLAG (Wako), anti-V5 (Invitrogen), and anti-HA (Sigma-Aldrich, St. Louis, MO, USA). After incubation, the primary antibodies were detected using secondary antibodies (anti-mouse IgG, rabbit IgG or rat IgG; GE Healthcare) conjugated with horseradish peroxidase (HRP) and an ECL chemiluminescent detection system, Immuno Zeta (Wako), according to the manufacturer’s protocol.

### 2.12. Reverse Transcription-qPCR

Total RNA from the cells was isolated by use of ISOGEN II (Nippon Gene, Tokyo, Japan), and 1 μg of RNA was converted into cDNA using a High-Capacity RNA to cDNA Master Mix kit (Applied Biosystem, Waltham, MA, USA) according to the manufacturer’s guidelines. Real-time quantitative PCR (RT-qPCR) was performed using a SYBR qPCR mix (Nippon Gene). Fluorescence emitted by SYBR green was detected using an ABI PRISM 7500 sequence detector. β-actin was used as an internal control and to normalize the values. The experiments were performed in triplicate for all the samples. The primers used for RT-qPCR were as follows: (PHLPP1: Forward: ACACCGTGATTGCTCACTCC, Reverse: TTCCAGTCAGGTCTAGCTCC; NEDD4-1: Forward: CCTAAAGGCTGGGAAGTCCG, Reverse: GATCTTCCCAGGTGGTGGTT; NEDD4-2: Forward: ATGACTCGGCTTCTCAGCAC, Reverse: TCCGGTTGTTGTGGTTGACA; SMURF1: Forward: CAACAGTCCAGGGCCAAGTT, Reverse: AAGGTCTCTTGGTATCCTGGGG; SMURF2: Forward: TGGGAAGAAAGGAGAACCGC, Reverse: ATATTCGGATGCCGGTCGTG; HECW2: Forward: CACCAGGGCAAGGCATTCT, Reverse: GTTGCCTTGTCAGGTGTTGC; and Human β-actin: Forward: GCACTCTTCCAGCCTTCCTT, Reverse: CGTACAGGTCTTTGCGGATG).

### 2.13. Co-Immunoprecipitation (Co-IP)

HEK-293T cells were transfected with the indicated plasmids, and lysates were prepared using a TNE buffer with a protease and phosphatase inhibitor cocktail. Approximately 10% of the lysates were used as input samples. The rest of the lysates were incubated with the indicated antibodies at 4 °C overnight, after which protein G or A beads were added to each sample and incubated at 4 °C for 50 min with end-over-end rotation. Then, the beads were washed with a lysis buffer at least 3 times, after which a SDS sample buffer was added to the beads and the samples were boiled at 98 °C for 5 min.

### 2.14. Xenograft Mouse Model for Tumorigenicity Studies

All in vivo experiments with animal models were approved and performed according to the guidelines of the ethics committee of the University of Tsukuba, Japan. Twenty-four female BALB-c/nude mice (4–5 weeks old, 15–20 g in weight) were purchased from CLEA, Tokyo, Japan and separated into four groups randomly before injection of the cancer cells. PHLPP1-KD and control cells from MDA-MB-231 cells (both in parental and in TMEPAI-KO cells) were resuspended in serum-free DMEM. Approximately 1 × 10^6^ cells in 100 μL DMEM with an equal amount of Matrigel were injected into each mouse subcutaneously. Six weeks after the injection, the mice were euthanized, and the excised tumors were weighed. A part of the tumor tissues was fixed with paraformaldehyde for immunochemical staining, and the remaining multiple tumors were combined and used for protein extraction for Western blot analysis.

### 2.15. Immunohistochemical Staining

Approximately 3-μm-thick tumor tissue slices were immunohistochemically analyzed. The tissue sections were deparaffinized with xylene and rehydrated with a descending order of ethanol solutions. For antigen retrieval, the slices were soaked in citrate buffer and heated at 121 °C for 20 min. Then, the slices were blocked with blocking reagent for 1 h, and subsequently incubated with Ki-67 primary antibody at 4 °C overnight. The primary antibody was detected by use of HRP-labelled polymer secondary antibody (EnVision, DAKO, Glostrup, Danmark) with DAB solution; the slices were then counterstained with hematoxylin and photographs were taken at different magnifications with an Olympus microscope, Olympus, Tokyo, Japan.

### 2.16. Statistical Analysis

All the quantitative analytic data are indicated as the means ± standard deviations (SDs) of three independent experiments. Differences between the parental and TMEPAI KO cells were determined using one-way analysis of variance (ANOVA). An unpaired *t* test was used to compare the control and test groups. Probability values less than 0.05 were considered significant. TCGA gene sets were calculated using the Fisher exact test, and survival analyses were performed using GraphPad Prism 7.

## 3. Results

### 3.1. TMEPAI Promotes Colony Formation, Tumor Sphere Formation, and Cell Migration in TNBC Cells

To examine the role of TMEPAI in TNBC cells, we previously generated TMEPAI-KO TNBC sublines using the CRISPR/Cas9 gene-editing technique [28,29]. In accordance with our previous result, TMEPAI-KO cells showed a remarkable reduction in clonogenicity as compared with the parental cells using two TNBC cell lines, Hs578T and BT-549 (Figure 1A,B), although no proliferation differences were found between parental and TMEPAI KO cells in ordinary monolayer cell culture condition (Appendix A). The tumor sphere formation assay measures anchorage-independent growth ability, which is closely related with in vivo tumorigenic activity. TMEPAI-KO cells formed significantly fewer numbers of large tumor spheres (≥100 and 75 µm in diameter in Hs578T and BT-549 cells, respectively) than did the parental cells (Figure 1C,D). To determine the effects of TMEPAI on cell motility, we carried out a transwell cell migration assay using MDA-MB-231, an invasive subtype of TNBC cells, and found that the TMEPAI-KO cells exhibited significantly less migration ability than that of the parental cells (Figure 1E). A consistent result was also found on wound healing assay (Appendix A). These results collectively indicated a critical oncogenic role of TMEPAI in TNBC cells.

### 3.2. Correlation of High Expression of TMEPAI and High Levels of AKT Ser473 Phosphorylation among TCGA Invasive Breast Cancer Cases (BRCA)

It was previously reported that TMEPAI expression is increased in TNBC cells and the high expression of TMEPAI is correlated with poor prognosis of ER/PR negative and lymph node positive breast cancer patients [12]. These studies combined with our previous results inspired us to further investigate the underlying molecular mechanism of TMEPAI in the progression of TNBC.

To address that, we performed gene set enrichment analysis (GSEA) of 1212 RNAseq samples from 1093 TCGA invasive breast cancer cases (BRCA), using the WikiPathways database to identify the top biological pathways enriched in TMEPAI.High samples as compared with the TMEPAI.Low samples (303 samples each, defined as described in the methods). We observed a repeated appearance of biological pathways related to the signatures of TGF-β signaling and PI3K/AKT signaling among the top biological pathways with the highest normalized enrichment score (NES), implying the importance of the association between TMEPAI status and these two signaling pathways (Figure 2A–C). Previously, we investigated the role of TMEPAI in negative regulation of TGF-β signaling (20). Hence, we here focused on the regulation of PI3K/AKT signaling.

To examine the association between alteration in mRNA expression of TMEPAI and changes in phosphorylation of AKT, we filtered 771 TCGA invasive breast cancer cases (BRCA) that had matched single mRNAseq and reverse phase protein array (RPPA) samples. In this filtered list, arranged in descending order by RSEM-normalized RNAseq values of TMEPAI, the AKT Ser473 phosphorylation and AKT Thr308 phosphorylation levels were significantly higher in the TMEPAI.High samples than in the TMEPAI.Low samples (193 samples each, defined as described in the methods) (Figure 2D,E, respectively). Since TMEPAI is known to be involved in the degradation of PTEN [12], which negatively regulates AKT phosphorylation at Thr308, we anticipated that TMEPAI probably participates in the activation of AKT through the phosphorylation at Ser473 by an alternative mechanism.

### 3.3. TMEPAI Promotes PI3K/AKT Signaling through Phosphorylation of Ser473 of AKT by Suppressing PHLPP1

To confirm the relationship between TMEPAI and phosphorylation of AKT at Ser473, we examined the levels of AKT Ser473 phosphorylation in our TMEPAI-KO TNBC cell lines. Since the expression of TMEPAI was relatively low in ordinary cell culture conditions, we treated the cells with TGF-b to induce TMEPAI at high levels. As a result, TMEPAI expression was proportional to AKT Ser473 phosphorylation without a change in the total AKT protein levels. Interestingly, we also found that the amount of PHLPP1 was remarkably increased in the TMEPAI-KO cells as compared with the parental cells (Figure 3A,B). Consistent results were also observed in another TNBC cell line, MDA-MB-231 (Appendix A). As PHLPP1 is a negative regulator of PI3K/AKT signaling, we examined whether the decreased AKT Ser473 phosphorylation was due to the increase in PHLPP1 in the TMEPAI-KO cells. Successful PHLPP1 KD was achieved by siRNA in both the parental and the TMEPAI-KO cells. As expected, KD of PHLPP1 from the TMEPAI-KO cells rescued more than 70% of the AKT Ser473 phosphorylation reduced by TMEPAI KO (Figure 3C,D). We also found similar results using shPHLPP1 in MDA-MB-231 cell lines (Appendix A). These data clearly demonstrated that TMEPAI suppresses the amount of PHLPP1 to promote AKT Ser473 phosphorylation.

### 3.4. Knockdown of PHLPP1 Rescues Colony Formation, Tumor Sphere Formation, and Cell Migration in TMEPAI-KO Cells

PI3K/AKT signaling is well known to be implicated in tumorigenic activities in many types of cancer, and it is already reported that PHLPP1 has a tumor-suppressive role through the negative regulation of PI3K/AKT signaling. Thus, TMEPAI may promote TNBC tumorigenesis depending on PHLPP1 function. To further substantiate our hypothesis, we knocked down PHLPP1 and carried out in vitro tumorigenic assays. Accordingly, the reduced colony-forming ability of TMEPAI-KO cells was significantly recovered by PHLPP1 KD (Figure 4A,B). KD of PHLPP1 also partially rescued the number of large tumor spheres (Figure 4C,D) and the cell migration in TMEPAI-KO cells (Figure 4E). Furthermore, to confirm the involvement of AKT signaling in in vitro tumorigenicity, we treated the TNBC cells with AKT inhibitor and found that AKT inhibitor dramatically suppressed AKT Ser473 phosphorylation (Appendix A) and tumor-sphere formation (Appendix A). Collectively, these results indicate an essential role of dephosphorylation of AKT catalyzed by PHLPP1 in suppression of in vitro tumorigenicity in TMEPAI-KO TNBC cells.

### 3.5. Knockdown of PHLPP1 Promotes In Vivo Tumor Formation in a Mouse Xenograft Model

Next, we aimed to investigate the effect of PHLPP1 KD on xenograft tumor formation. We established stable PHLLP1 KD cell lines from MDA-MB-231 parental and TMEPAI KO cells using a PHLPP1 shRNA (shPHLPP1) expression system and injected the cells subcutaneously into immunodeficient mice. The growth of tumors was markedly increased in the PHLPP1-KD cells as compared with their respective control cells (Figure 5A,B and Appendix A). Moreover, Western blot analytic data from the tumor tissue samples confirmed the upregulated Ser473 phosphorylation of AKT due to the PHLPP1 KD (Figure 5C). Furthermore, the immunohistochemical analysis clearly demonstrated that PHLPP1 KD increased the number of Ki-67-positive cells (Figure 5D,E). These results suggest that decreased AKT Ser473 phosphorylation by PHLPP1 suppresses in vivo tumor growth and that TMEPAI antagonizes these PHLPP1 effects in xenograft tumors.

### 3.6. PY Motifs of TMEPAI Mediate Downregulation of PHLPP1 to Facilitate AKT Ser473 Phosphorylation

As TMEPAI has known functional motifs, i.e., a SIM and double PY motifs, we aimed to confirm which motif of TMEPAI is responsible for the downregulation of PHLPP1. To examine this, TMEPAI isoform A wild-type (WT) and each motif mutant were re-expressed in Hs578T TMEPAI-KO cells (Figure 6A). The double PY motif mutant could not suppress the amount of PHLPP1 protein that the TMEPAI WT and SIM mutant did. Additionally, re-expression of the double PY mutant could not increase the phosphorylation of AKT Ser473 in TMEPAI-KO cells (Figure 6B). The colony-forming ability of the re-expressed cells also corroborated the above observation (Figure 6C). Therefore, all these findings indicate that TMEPAI downregulates PHLPP1 through PY motifs and assists AKT activation.

### 3.7. NEDD4-2 Mediates TMEPAI-Induced Proteasomal Degradation of PHLPP1

To investigate the mechanism by which double PY motifs of TMEPAI are responsible for the suppression of PHLPP1, we first checked the mRNA levels of PHLPP1, both in TMEPAI parental and in KO cells, and found that there were no changes in the PHLPP1 mRNA levels (Appendix A). Additionally, MG-132, a proteasome inhibitor, increased the amount of PHLPP1 protein along with the duration of treatment (Appendix A), suggesting that PHLPP1 may be regulated by proteasomal degradation but not by transcriptional regulation. The PY motif is known to be an interaction site for WW domain-containing protein, and it has been reported that TMEPAI interacts with several HECT E3 ubiquitin ligases [19]. Therefore, we checked the mRNA levels of different HECT E3 ligases and found that NEDD4-2 and SMURF1 were highly expressed (Appendix A), and protein expression of NEDD4-2 and SMURF1 was further confirmed in TNBC cells (Appendix A). To confirm the effects of NEDD4-2 and SMURF1 on PHLPP1 regulation, we knocked down NEDD4-2 or SMURF1 in Hs578T cells and found that NEDD4-2 KD recovered the amount of PHLPP1 (Figure 7A), but SMURF1 KD did not (Figure 7B). These observations indicated that NEDD4-2 is an important E3 ligase that regulates PHLPP1.

To delineate the molecular mechanism of TMEPAI that supports NEDD4-2 dependent PHLPP1 degradation, we examined the protein interactions among them. The co-immunoprecipitation (IP) experiments demonstrated that TMEPAI (WT, a SIM 4A mutant, and a SIM P186A mutant) binds with NEDD4-2; however, a double PY motif mutant was unable to bind with NEDD4-2 (Figure 7C). We also confirmed the interaction between NEDD4-2 and PHLPP1 (Figure 7D). However, we could not detect visible interaction between TMEPAI and PHLPP1 (Figure 7E). Next, we examined whether TMEPAI affects the complex formation of NEDD4-2 with PHLPP1. Indeed, the interaction between PHLPP1 and NEDD4-2 was increased in the presence of TMEPAI (Figure 7F). To further confirm the involvement of TMEPAI and NEDD4-2 in the proteasomal degradation of PHLPP1, a ubiquitination assay was performed in the presence of a proteasomal inhibitor. Under this condition, polyubiquitinated PHLPP1 was increased by NEDD4-2 and further enhanced in the co-expression of TMEPAI (Figure 7G). These data suggested that the interaction of TMEPAI with NEDD4-2 activates the complex formation with PHLPP1 and enhances PHLPP1 polyubiquitination, which leads to the proteasomal degradation of PHLPP1.

## 4. Discussion

Multiple integrated signaling networks tightly regulate tissue homeostasis and cellular functions. Imbalance among these intracellular signaling components leads to various life-limiting diseases, including cancer. During cancer progression and metastasis, various oncogene products are upregulated or constitutively activated, which in turn accelerates cancer-promoting signaling. It has already been reported that TMEPAI expression is highly upregulated in different types of cancer [21,27,34,35,36] and its expression is inversely correlated with patients’ survival rates [22,37]. Our current study demonstrated that TMEPAI KO decreased tumorigenic activities, such as tumorigenic proliferation and migration, when compared with parental TNBC cells. Thus, TGF-b-induced TMEPAI is probably implicated in malignant progression of TNBC, in accordance with previous studies on many types of cancer cells [27,28,29].

We previously reported that TMEPAI is strongly induced by TGF-β signaling in cooperation with EGF signaling [24]. TGF-β was originally identified from the supernatants of murine sarcoma virus-transformed fibroblasts as a factor that induces anchorage-independent growth in soft agar in the presence of TGF-α, a member of the EGF family. Although many researchers have been trying to investigate the TGF-β target genes responsible for the malignant transformation, here we have disclosed a unique oncogenic function of a TGF-β-inducible protein, TMEPAI, in triple-negative breast cancer cells.

Many studies revealed that PI3K/AKT signaling is commonly activated in most types of cancer [38,39] and has essential functions in the promotion of cell survival, cell cycle progression, and tumor growth [40,41]. In this study, we showed that TMEPAI-KO cells significantly reduced AKT Ser473 phosphorylation levels in connection with enhanced expression of PHLPP1 in comparison with parental cells. Therefore, TMEPAI positively regulates breast cancer progression by enhancing AKT Ser473 phosphorylation through the degradation of PHLPP1. The significance was further confirmed by TCGA gene set analysis, which showed a positive correlation between TMEPAI mRNA levels and phosphorylation levels of AKT Ser473 in invasive breast cancer patients.

It is well known that AKT has two important phosphorylation sites, Thr308 and Ser473 [42,43], and that phosphorylation of Ser473 is more than 80% responsible for full activation of AKT [10,11]. Although the upstream pathway of AKT is frequently activated in cancer cells, owing to receptor and PI3K activation or PTEN inactivation, PHLPP1 suppresses AKT signaling by specific dephosphorylation of AKT Ser473 [10,13,14,15,16]. Several studies reported that inactivation of AKT Ser473 by PHLPP1 suppresses tumorigenicity in cancer cells [13,44]. However, PHLPP1 inactive mutation in cancer cells has rarely been reported. Our study is the first to show that the oncogenic protein TMEPAI suppresses PHLPP1 and increases tumorigenic activities through AKT Ser473 phosphorylation. Although PHLPP1 KD did not bring about full recovery of tumor growth in TMEPAI-KO MDA-MB 231 cells, we hypothesize that it is probably due to the tumorigenic function of TMEPAI through SIM-mediated signal regulation [29] and/or suppression of PTEN [12].

Many researchers have been striving to explore an anticancer treatment targeting PI3K/AKT signaling; for example, apitolisib (GDC0980), dactosilib (BEZ235), and LY294002 (SF1101) are FDA-approved PI3K/AKT inhibitors [45]. However, PI3K/AKT signaling is essential for normal cells, and it seems that targeting the cardinal molecules of PI3K/AKT signaling may have severe side effects. Therefore, targeting the driver molecules responsible for hyperactivation of PI3K/AKT signaling specifically activated in cancer cells will be important to develop effective anticancer therapy with less adverse effects. TMEPAI is now considered to be just such an oncogenic molecule that is highly expressed in cancer cells and activates AKT independently from PI3K active mutations and loss of PTEN function.

As TMEPAI has a long intracellular domain, with one SIM and two PY motifs [20], it is important to know which motif of TMEPAI is responsible for the downregulation of PHLPP1and how TMEPAI promotes PI3K/AKT signaling. Our data suggest that the mutations in TMEPAI PY motifs diminished the inhibitory function on PHLPP1. Previous studies also revealed that TMEPAI PY motifs are the binding interface to the WW domain containing proteins such as NEDD4, and that they downregulate several target proteins including PTEN and androgen receptors [12,46,47]. Consistently, TMEPAI interacts with NEDD4-2, and this interaction enhances NEDD4-2’s binding to PHLPP1 and promotes ubiquitination and the subsequent proteasomal degradation of PHLPP1. Although further investigation is required to clarify the detailed mechanism, TMEPAI may change the binding ability of NEDD4-2 to PHLPP1 by conformational changes or another modification. In addition, membrane-bound TMEPAI may recruit NEDD4-2 close to the plasma and endosomal membranes where PHLPP1 works.

## 5. Conclusions

In summary, TMEPAI promotes TNBC progression by activating AKT Ser473 phosphorylation through downregulation of PHLPP1. Knockdown of PHLPP1 in TMEPAI-KO cells significantly restored tumorigenic abilities by enhancing AKT Ser473 phosphorylation, thereby delineating the opposite functions of TMEPAI and PHLPP1. TMEPAI binds to NEDD4-2 through its double PY motifs, enhances NEDD4-2 binding to PHLPP1, and activates ubiquitination and proteasomal degradation of PHLPP1. Therefore, an understanding of the oncogenic role of TMEPAI may be conducive to the development of novel TMEPAI-targeted therapy for TNBC in the future.

## Figures and Tables

**Figure 1 cancers-13-04934-f001:**
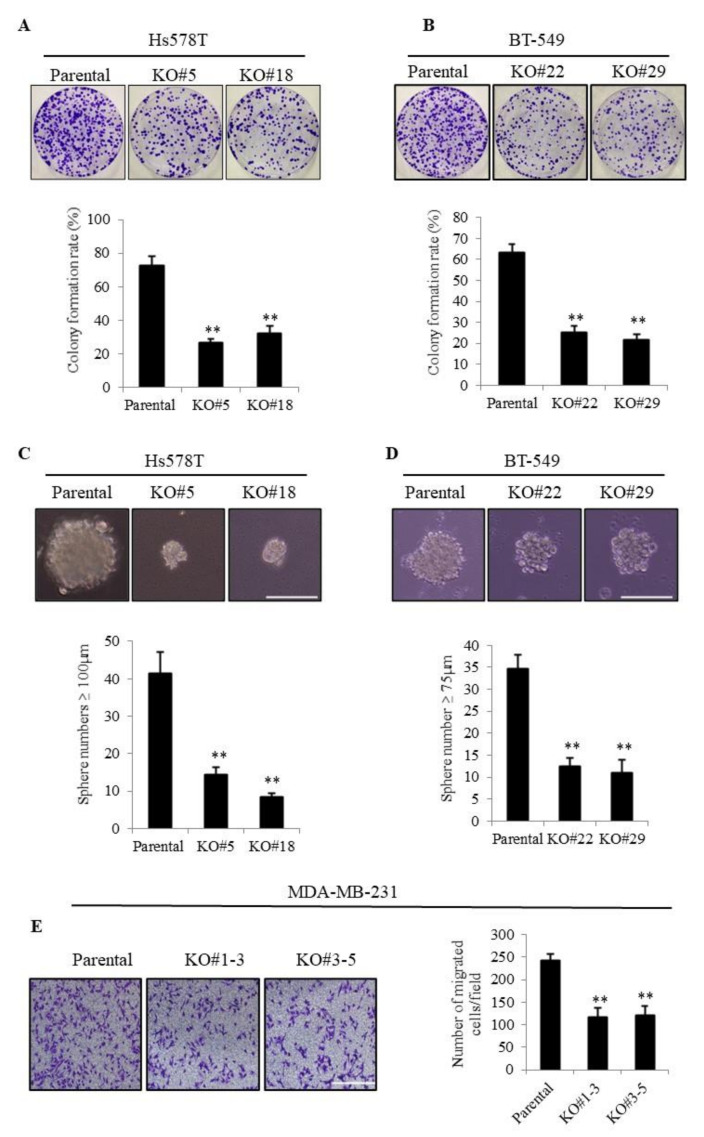
Roles of TMEPAI in breast cancer cell proliferation, tumor sphere formation, and cell migration. (**A**,**B**), Colony formation assays were performed using parental and TMEPAI-KO TNBC cell clones (Hs578T #5 and #18(**A**) and BT-549 #22 and #29 (**B**), respectively). Representative photographs of colonies and calculated colony formation rate. (**C**,**D**), Representative tumor spheres of parental and TMEPAI-KO cells (Hs578T (**C**) and BT-549 (**D**), respectively). The scale bar indicates 100 mm. The bar charts depict the numbers of large spheres formed by the parental and TMEPAI-KO cells (Hs578T and BT-549, respectively). (**E**), The migration ability of parental and TMEPAI-KO cells (MDA-MB-231) was determined by a transwell cell migration assay and photographs were taken to calculate the migrated cell numbers. The scale bar indicates 100 mm. The values presented here are the means ± SDs of 3 independent experiments. The bars with asterisks indicate significant differences between the parental cells at *p* ≤ 0.01 (**).

**Figure 2 cancers-13-04934-f002:**
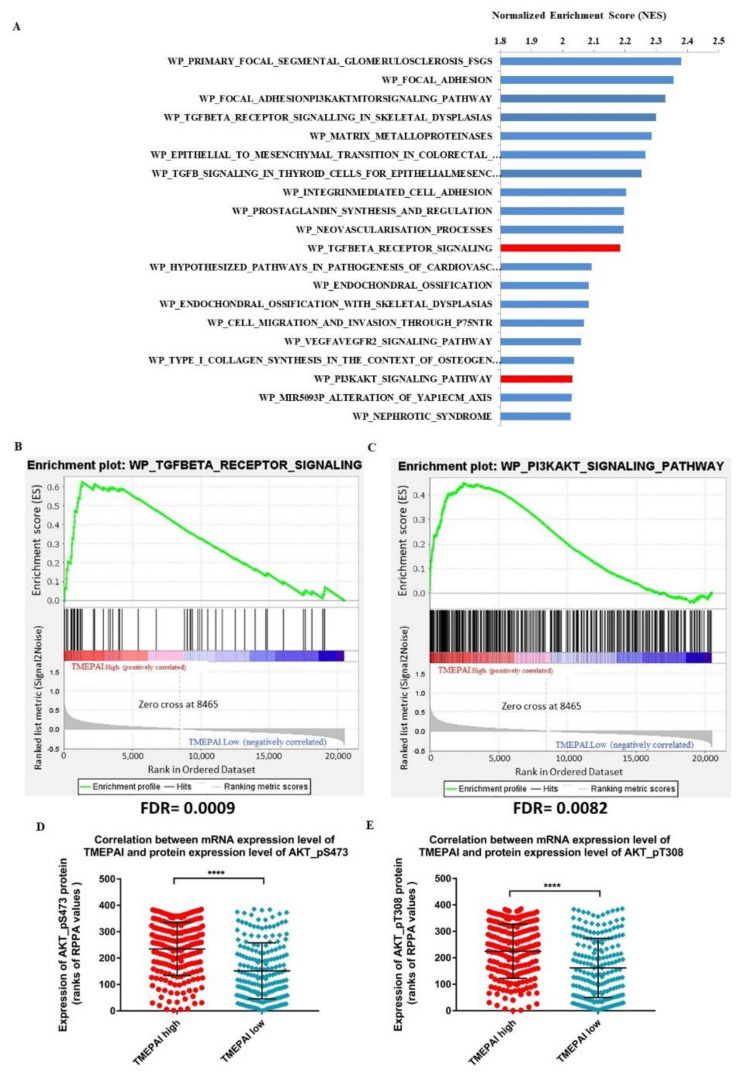
Association between altered expression of TMEPAI mRNA and AKT Ser473 or AKT Thr308 in TCGA-invasive breast cancer cases. (**A**), Top biological pathways with the highest normalized enrichment score (NES) identified on gene set enrichment analysis (GSEA) of TMEPAI.High samples (303) compared with TMEPAI.Low samples (303) in TCGA invasive breast cancer cases (BRCA) at FDR q-value < 0.05. (**B**,**C**), Enrichment plot of signatures of the TGF-β receptor signaling (**B**) and PI3K/AKT signaling (**C**) pathways showing the profile of the running enrichment score and the positions of gene set members on the rank-ordered list of upregulated genes in TMEPAI.High TCGA invasive breast cancer cases (BRCA). (**D**,**E**), Comparison of ranks of reverse phase protein array values (RPPA) of AKT Ser473 (**D**) or AKT Thr308 (**E**) between TMEPAI.High and TMEPAI.Low samples in cases that have matched single mRNAseq and RPPA samples (193 samples each); Mann–Whitney test, **** *p* value < 0.0001, figures show Mean ranks and standard deviation.

**Figure 3 cancers-13-04934-f003:**
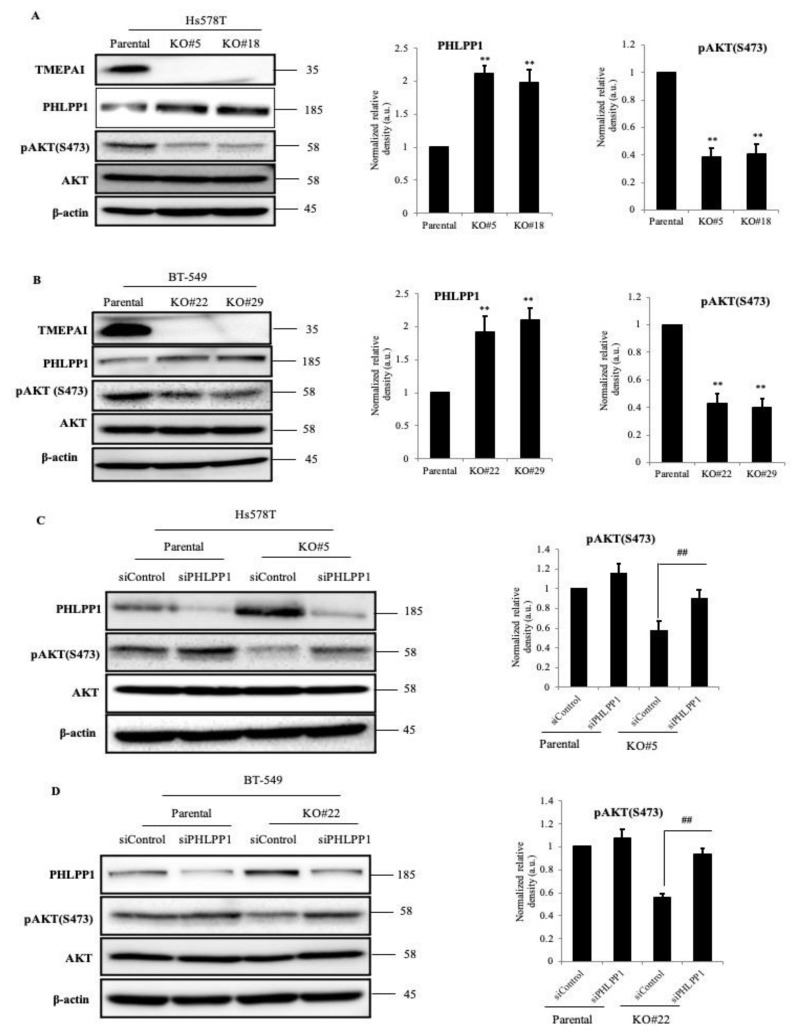
Effects of TMEPAI on AKT signal activation. (**A**,**B**), Lysates from TGF-β stimulated parental and TMEPAI-KO cells [Hs578T (**A**) and BT-549 (**B**)] were subjected to Western blot analysis to detect the amounts of TMEPAI, PHLPP1, pAKT(S473), and AKT. The bar charts express the relative PHLPP1 and pAKT band intensity normalized to the corresponding β-actin band intensity. (**C**,**D**), PHLPP1-knockdown efficiency was confirmed and the amounts of pAKT(S473) and AKT were detected (Hs578T (**C**) and BT-549 (**D**), respectively) by means of Western blot analysis. β-actin was used as the loading control. The bar graphs depict the relative amount of pAKT (S473) in the control and PHLPP1 KD cells. The values presented here are the means ± SDs of 3 independent experiments. The bars with asterisks indicate significant difference from the parental vs. the TMEPAI-KO cells at *p* ≤ 0.01 (**) and from the siControl vs. the siPHLPP1 of TMEPAI-KO cells at *p* ≤ 0.01 (##). The uncropped Western Blot images can be found in Appendix A.

**Figure 4 cancers-13-04934-f004:**
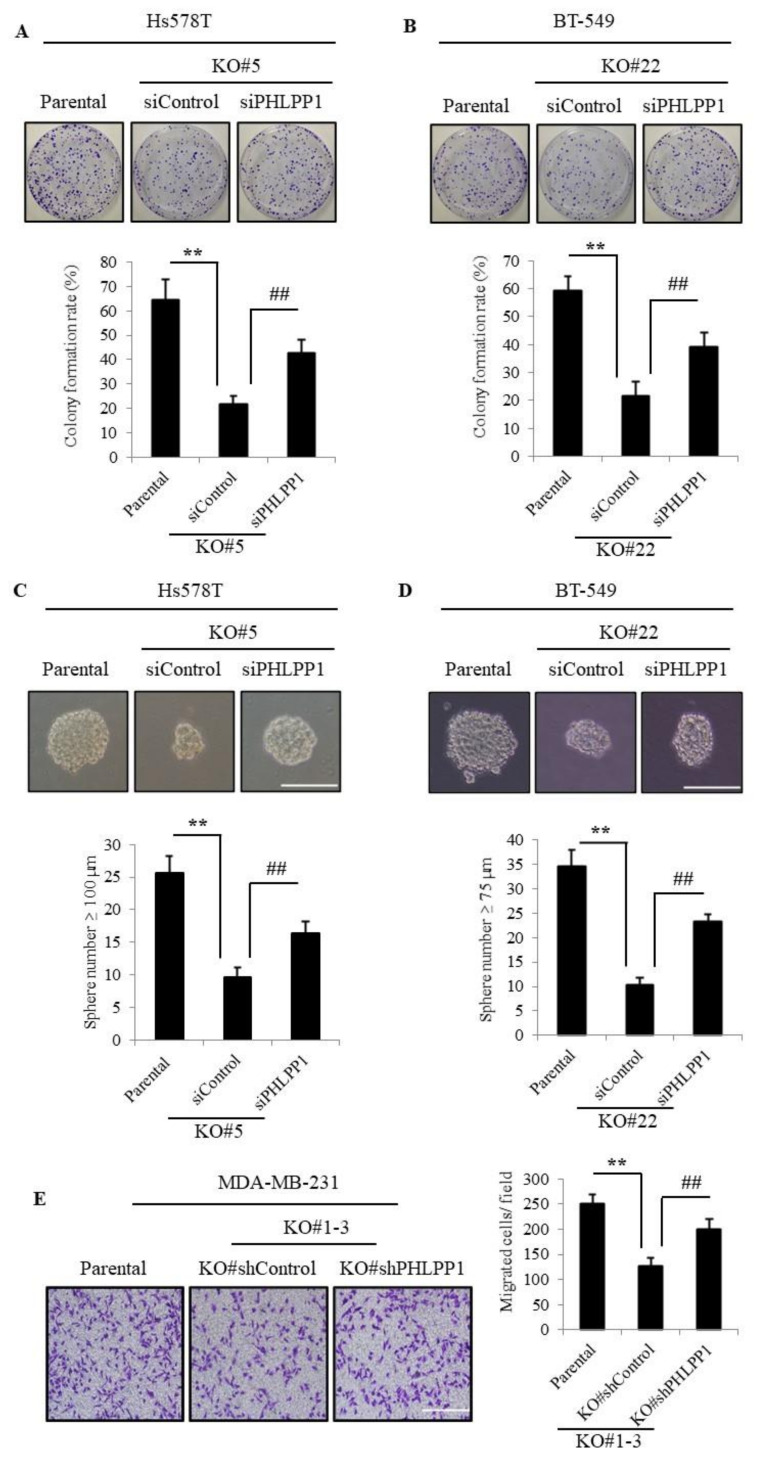
Knockdown of PHLPP1 can rescue colony numbers, tumor sphere growth, and cell migration in TMEPAI-KO breast cancer cells. (**A**,**B**), Colonies of PHLPP1-knockdown cells (Hs578T (**A**) and BT-549 (**B**), respectively) and the colony formation rates are shown in the bar graphs. (**C**,**D**), Tumor sphere-forming ability of PHLPP1-knockdown cells (Hs578T (**C**) and BT-549 (**D**)). The scale bar indicates 100 mm. (**E**) Migratory ability of PHLPP1-KD and control cells. The representative photos are shown, The scale bar indicates 100 mm. The values presented here are the means ± SDs of 3 independent experiments. The bars with asterisks indicate significant difference from the parental vs. the TMEPAI-KO cells at *p* ≤ 0.01 (**) and from the siControl (#EV) vs. the siPHLPP1(sh#PHLPP1) of the TMEPAI-KO cells at *p* ≤ 0.01 (##).

**Figure 5 cancers-13-04934-f005:**
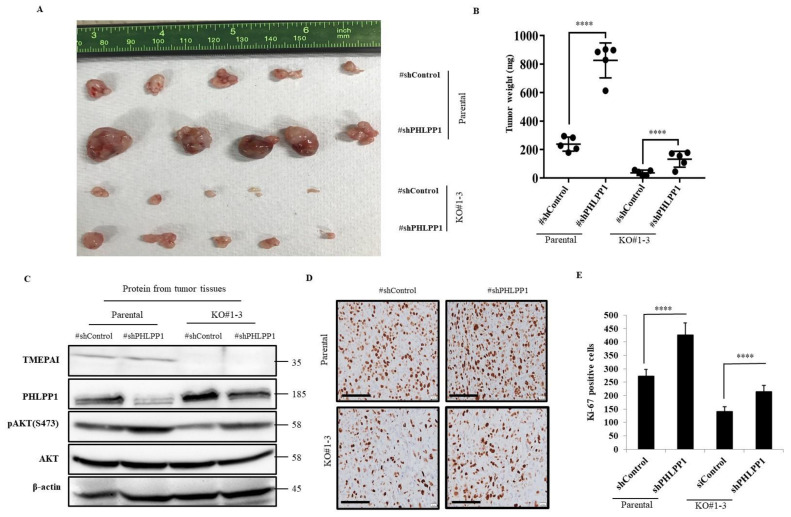
Knockdown of PHLPP1 promotes xenograft tumor formation in both parental and TMEPAI KO cells. (**A**), Macroscopic view of the tumors. PHLPP1-knockdown cells from parental and TMEPAI-KO MDA-MB-231 cells (1 × 10^6^) were subcutaneously injected into the nude mice. After one and a half months, the mice were sacrificed and the tumors collected from each group. (**B**), The collected tumors were weighed. (**C**), The tumor tissues were crushed with a homogenizer, and lysate samples were prepared for Western blot analysis. TMEPAI, PHLPP1, pAKT(S473), and total AKT were detected as indicated. β-actin was used as the loading control. (**D**,**E**), Representative pictures and calculated proliferative index (respectively) of tumor xenografts stained with anti Ki-67 antibody. The scale bar indicates 100 mm, Differences between each datum were evaluated using one-way ANOVA and the *t* test; *p* ≤ 0.001 (****). The uncropped Western Blot images can be found in Appendix A.

**Figure 6 cancers-13-04934-f006:**
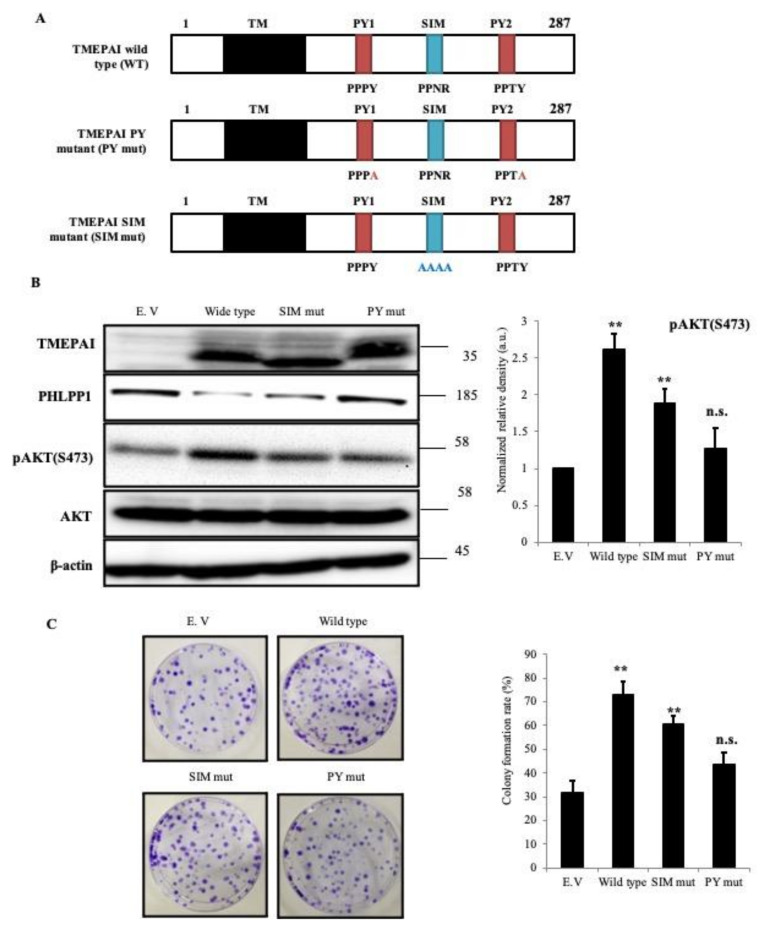
PY motifs of TMEPAI are required for downregulation of PHLPP1 to sustain AKT phosphorylation. (**A**), Schematic representation of mutations in SIM and PY motifs of human TMEPAI. (**B**), Lysates from TMEPAI re-expressed cells were subjected to Western blot analysis to detect TMEPAI, PHLPP1, pAKT(S473), and AKT as indicated. The bar chart depicts the relative densities of the pAKT(S473) bands. (**C**), The visualized colonies and their respective colony-forming rates are shown. The values presented here are the means ± SDs of 3 independent experiments; *p* ≤ 0.01 (**), where n.s. means not significant. The uncropped Western Blot images can be found in Appendix A.

**Figure 7 cancers-13-04934-f007:**
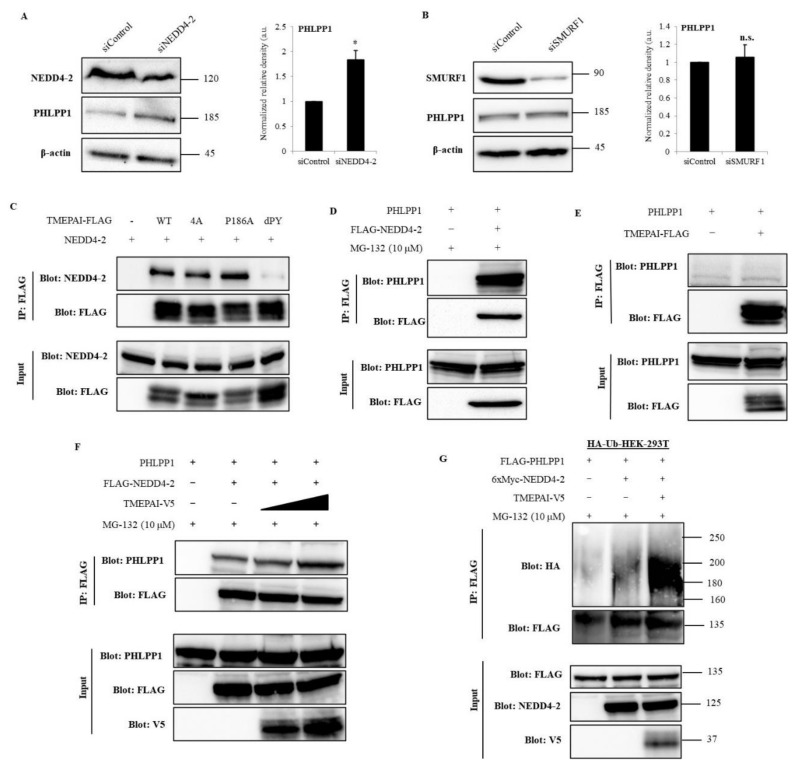
TMEPAI destabilizes PHLPP1 by enhancing the complex formation of NEDD4-2 and PHLPP1 and promotes PHLPP1 ubiquitination. (**A**,**B**), KD efficiency of NEDD4-2 (**A**) and SMURF1 (**B**) and their effects on PHLPP1 amounts in Hs578T cells. The values presented here are the means ± SDs of 3 independent experiments; *p* ≤ 0.05 (*) where n.s. means not significant. (**C**), Co-immunoprecipitation of TMEPAI and NEDD4-2. WT, wild type; 4A, a SIM 4A mutant; P186A, a SIM P186A mutant; dPY, a double PY mutant. (**D**), Co-immunoprecipitation of NEDD4-2 and PHLPP1 in the presence of MG132. (**E**), Co-immunoprecipitation of TMEPAI and PHLPP1 in the presence of MG132 showing no detectable interaction between TMEPAI and PHLPP1. (**F**), Co-precipitated PHLPP1 with FLAG-NEDD4-2 was detected with increasing amounts of TMEPAI-V5. (**G**), HEK-293T cells stably expressing HA-ubiquitin (HA-Ub-HEK-293T) were transfected with TMEPAI-V5, FLAG-PHLPP1, and 6xMyc-NEDD4-2, and ubiquitinated PHLPP1 was detected by immunoprecipitation of FLAG-PHLPP1 and blotting with anti-HA antibody. The cell lysates were directly subjected to Western blot analysis to detect the total amounts of the expressed proteins (Input). The uncropped Western Blot images can be found in Appendix A.

## Data Availability

No new data were created or analyzed in this study. Data sharing is not applicable to this article.

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
