# Peer review of "PMEPA1/TMEPAI Is a Unique Tumorigenic Activator of AKT Promoting Proteasomal Degradation of PHLPP1 in Triple-Negative Breast Cancer Cells"

_cancers, 2021, doi:10.3390/cancers13194934_

Round 1
Reviewer 1 Report
In this article the authors have identified TMEPAI as activator of AKT Ser473 in a triple negative breast cancer cell line.
Overall the study is interesting and it brings more light into the complex regulation of PI3K and AKT signalling. Nevertheless, I am lacking some link with TNBC besides the analysis of just one cell line and datasets. Have the authors checked the expression of TMEPAI in TNBC tumors and correlated it with patient's outcome?
In addition, to demonstrate the potential of translation into the clinics, have the authors tested PI3K, AKT or mTOR inhibitors or any of the approved drugs mentioned in 573 and 537, in the developed cell lines? It would also be interesting to include a drug assay on the xenograft models developed by the authors.
Author Response
Thank you for the constructive comments that helped us improved our manuscript. Here is our response to each comment.
Overall the study is interesting and it brings more light into the complex regulation of PI3K and AKT signalling. Nevertheless, I am lacking some link with TNBC besides the analysis of just one cell line and datasets. Have the authors checked the expression of TMEPAI in TNBC tumors and correlated it with patient's outcome?
In this manuscript, we used three triple negative breast cancer cell lines, Hs578T, BT-549, and MDA-MB-231, and all cell lines similarly showed that TMEPAI regulated PHLPP1 degradation and AKT Ser473 phosphorylation which led to the tumorigenic phenotype. Unfortunately, we do not have easy access to clinical specimens and data. However, the correlation of TMEPAI expression and clinical outcome of TNBC patients was previously described in Singha P.K. et.al., Genes Cancer 2014, 5(9-10), page 320-336 (reference no. 12). In this paper, they showed that TMEPAI expression was increased in TNBC cells and TMEPAI high group of ER/PR negative and lymph node positive breast cancer patients was positively correlated with shorter recurrence-free survival. We cited their paper and described in the text. (line 335-339)
In addition, to demonstrate the potential of translation into the clinics, have the authors tested PI3K, AKT or mTOR inhibitors or any of the approved drugs mentioned in 573 and 537, in the developed cell lines? It would also be interesting to include a drug assay on the xenograft models developed by the authors.
We carried out sphere formation assay using allosteric AKT inhibitor (MK-2206). The result showed that AKT inhibitor suppressed the AKT Ser473 phosphorylation and sphere forming ability in TNBC cell lines (Supplementary Fig. S3) (line 413-416). Due to time limitation, we could not perform xenograft experiment using the above AKT inhibitor. Additionally, FDA-approved PI3K/AKT inhibitors are currently in use for many types of cancer including breast cancer.
As we mentioned in discussion (line 561-570), we believed that TMEPAI has a potential to be a molecular target for cancer therapy with fewer side effects compared with PI3K/AKT inhibitors due to its high expression in cancer cells. Now we are screening compounds against TMEPAI and wish to report the anti-tumorigenic effect in the future.
Reviewer 2 Report
The present manuscript is well written, focused on the role of PMEPA1/TMEPAI in regulating the Ser473 phosphorylation of AKT in Triple-Negative Breast Cancer Cells. The manuscript would be strengthened by addressing the points below.
My comments:
- In figure 1, A and B. Authors showed knockout of TMEPAI effects the proliferation of breast cancer cells. However, authors need to show this phenotype in other methods such as MTT or XTT assay. Similarly in cell migration assay, show the phenotype other methods such as wound healing assay.
- Figure 5, A. authors need to show the tumor growth curve along with a picture of A.
- Figure 5, C. authors need to show the western blot data from a couple of tumor tissues of the same condition in the same blot.
- Figure 5, D. Authors need to show the representative pictures of HIC of Ki-67, along with quantitative data.
Author Response
Thank you for the constructive comments that helped us improved our manuscript. Here is our response to each comment.
- In figure 1, A and B. Authors showed knockout of TMEPAI effects the proliferation of breast cancer cells. However, authors need to show this phenotype in other methods such as MTT or XTT assay. Similarly in cell migration assay, show the phenotype other methods such as wound healing assay.
We carried out MTA assay and would healing assays. The results are shown in Supplementary Fig. S1A-D (line 308-312). In accordance with our previous result, there was no difference in cellular proliferation in parental and TMEPAI KO cells in ordinary monolayer culture condition. However, TMEPAI KO clearly reduced the colony and sphere forming abilities as well as cell migration ability. We added cell proliferation assay and would healing assay procedures in the method section (line 181-185, 208-215)
- Figure 5, A. authors need to show the tumor growth curve along with a picture of A.
We added a tumor growth curve data as shown in Supplementary Fig. S4 (line 436).
- Figure 5, C. authors need to show the western blot data from a couple of tumor tissues of the same condition in the same blot.
Because TMEPAI KO cells generate smaller sized tumors in vivo as shown in Fig. 5A, we lysed multiple tumors to obtain enough proteins for western blot analysis. Unfortunately we cannot provide the western blot that reviewer suggested. However, the result of Fig. 5C shows the average expression of the protein from the couple of tumor tissues in each condition. We described this in the method (line 283-284).
- Figure 5, D. Authors need to show the representative pictures of HIC of Ki-67, along with quantitative data.
We added the representative immunohistochemical pictures of Ki-67 as Fig.5D (line 439).
Round 2
Reviewer 1 Report
The authors have answered satisfactory the addressed questions.
Reviewer 2 Report
Authors have addressed all comments, at their best. the manuscript is improvised after revision, the present manuscript is in acceptable form.